1

# A global daily seamless 9-km Vegetation Optical Depth (VOD) product from 2010 to 2021

А

| 3  | Die Hu", Yuan Wang", Han Jing", Linwei Yue", Qiang Zhang", Lei Fan", Qiangqiang                |
|----|------------------------------------------------------------------------------------------------|
| 4  | Yuan <sup>†a,f,g</sup> , Huanfeng Shen <sup>h</sup> , and Liangpei Zhang <sup>i</sup>          |
| F  | <sup>a</sup> School of Geodesy and Geomatics, Wuhan University, Wuhan, China                   |
| 5  | behool of Atmospheric Dhusica, Nanjing University of Information Science and                   |
| 6  | School of Atmospheric Physics, Nanjing University of finormation Science and                   |
| 7  | Technology, Nanjing, China                                                                     |
| 8  | <sup>c</sup> School of Geography and Information Engineering, China University of Geosciences, |
| 9  | Wuhan, China                                                                                   |
| 10 | <sup>d</sup> Center of Hyperspectral Imaging in Remote Sensing (CHIRS), Information Science    |
| 11 | and Technology College, Dalian Maritime University, Dalian, China                              |
| 12 | <sup>e</sup> Chongqing Jinfo Mountain Karst Ecosystem National Observation and Research        |
| 13 | Station, School of Geographical Sciences, Southwest University, Chongqing, China               |
| 14 | <sup>f</sup> Key Laboratory of Geospace Environment and Geodesy, Ministry of Education,        |
| 15 | Wuhan University, Wuhan, China                                                                 |
| 16 | <sup>g</sup> Key Laboratory of PolarEnvironment Monitoring and Public Governance, Ministry     |
| 17 | of Education, Wuhan University, Wuhan, China                                                   |
| 18 | <sup>h</sup> School of Resource and Environmental Science, Wuhan University, Wuhan, China      |
| 19 | <sup>i</sup> State Key Laboratory of Information Engineering in Surveying, Mapping, and        |
| 20 | Remote Sensing, Wuhan University, Wuhan, China                                                 |

Abstract. Vegetation optical depth (VOD) products provide information on vegetation water content 21 and correlate with vegetation growth status, which are closely related to the global water and carbon 22 cycles. The L-band signal penetrates deeper into the vegetation canopy than the higher frequency 23 bands used for many previous VOD retrievals. Currently, there are only two operational L-band 24 sensors aboard satellites, namely the SMOS satellite launched in 2010 and the SMAP satellite launched 25 in 2015. The former has the limitation of a low spatial resolution of only 25 km, while the latter 26 has improved the resolution to 9 km but has a shorter usable time range. Due to the influence 27 of sensor and atmospheric conditions, as well as the observation methods of polar-orbiting satellites 28 (such as scan gaps and observation revisit times), the daily data provided by both satellites suffer from 29 varying degrees of missing data. In summary, the existing L-VOD products suffer from the defects 30 of missing data and coarse resolution of historical data. There is few research on filling gaps and 31 reconstructing 9-km long-term data for L-VOD products. To solve this problem, our study depends 32 on a penalized least square regression based on three-dimensional discrete cosine transform to firstly 33 generate the seamless global daily L-VOD products. Subsequently, the non-local filtering idea is applied 34 to spatiotemporal fusion between high- and low-resolution data, resulting in a global daily seamless 35 9-km L-VOD product from 1 January 2010 to 31 July 2021. In order to validate the quality of the 36 products, time series validation and simulated missing regions validation are used for the reconstructed 37 data. The fusion products are validated both temporally and spatially, and also compared numerically 38 with the original 9-km data during the overlapping period. Results show that the seamless SMOS 39 (SMAP) dataset is evaluated with a coefficient of determination  $(R^2)$  of 0.855 (0.947), and root mean 40 squared error (RMSE) of 0.094 (0.073) for the simulated real missing masks. The temporal consistency 41 of the reconstructed daily L-VOD products is ensured with the original time-series distribution of valid 42

h

тт

<sup>\*</sup>Corresponding author at: leifan33@swu.edu.cn

 $<sup>^{\</sup>dagger}\mathrm{Corresponding}$  author at: qqyuan@sgg.whu.edu.cn

values. The spatial information of the fusion product and the original 9-km data in the overlapping
period is basically consistent (R<sup>2</sup>: 0.926-0.958, RMSE: 0.072-0.093, MAE: 0.047-0.064). The temporal
variations between the fusion product and the original product are largely synchronized. Our dataset
can provide timely vegetation information during natural disasters (e.g., floods, droughts, and forest
fires), supporting early disaster warning and real-time response. This dataset can be downloaded at
https://doi.org/10.5281/zenodo.13334757 (Hu et al., 2024).

Keywords: SMOS, SMAP, vegetation optical depth, seamless, global daily long-term, 9-km, spa tiotemporal fusion

### <sup>51</sup> 1 Introduction

Vegetation is a key factor in the energy, water, and carbon balance of the terrestrial surface, and 52 it is significantly affected by climate change and human activities (Frappart et al., 2020). Remote 53 sensing observations are commonly used to monitor vegetation dynamics and their temporal changes 54 from regional to global scales. Unlike traditional optically based technologies, microwave-frequency 55 sensors are almost unaffected by cloud cover (Moesinger et al., 2020). Microwave radiation passing 56 through the vegetation canopy undergoes an extinction effect, and the extent of this attenuation can 57 be observed by passive or active microwave satellites and is commonly referred to as the vegetation 58 optical depth (VOD) (Wigneron et al., 2017). It is increasingly used for monitoring various ecological 59 vegetation variables, which can provide frequent observations that are independent of atmospheric 60 conditions and cloud pollution. Soil moisture contribution is coupled with the effects of vegetation in 61 terms of absorption and scattering (Liu et al., 2012; Zhao et al., 2021), and water within the vegetation 62 attenuates the microwave signal (Yao et al., 2024), thus VOD is directly related to the vegetation 63 water content (VWC) (Dou et al., 2023; Fan et al., 2019; Holtzman et al., 2021; Konings et al., 2016). 64 VOD has been widely used in biomass monitoring, drought early warning, phenology analysis, and 65 other fields (Fan et al., 2023; Ferrazzoli et al., 2002; Kumar et al., 2021; Mialon et al., 2020; Moesinger 66 et al., 2022; Vaglio Laurin et al., 2020; Van Dijk et al., 2013; Vreugdenhil et al., 2022; Wigneron et al., 67 2020). VOD is affected by a number of factors, including density and type of vegetation and microwave 68 frequency. Many microwave remote sensing satellites provide VOD products in different microwave 69 bands (X-, Ku-, C-). However, as the frequency of the microwave signal decreases, resulting in longer 70 wavelengths, its ability to penetrate vegetation canopies increases (Frappart et al., 2020; Zhang et al., 71 2021a). Compared to VOD products in other bands, the low-frequency microwave product L-VOD 72 correlates better with VWC and biomass (Brandt et al., 2018; Cui et al., 2023; Unterholzner, 2023). 73 Currently, only SMOS and SMAP satellites provide VOD data based on the L-band, and both are 74 satellites targeting the monitoring of soil moisture (SM) and VWC (Wigneron et al., 2017). 75

The Soil Moisture and Ocean Salinity (SMOS) mission is to monitor the brightness temperature 76 of microwave radiation at the earth's surface, launched by the European Space Agency (ESA) in 2009 77 (Kerr et al., 2001, 2010). SMOS carries a passive microwave radiometer that can acquire data without 78 emitting microwave signals by using microwave signals naturally radiated from the earth's surface. 79 Currently, there are three main physically based SMOS L-VOD retrieval methods (Wigneron et al., 80 2021), respectively SMOS L2 (Kerr et al., 2012), SMOS L3 (Al Bitar et al., 2017), and SMOS-IC 81 (Fernandez-Moran et al., 2017). These algorithms are all based on the L-band Microwave Emission of 82 the Biosphere (L-MEB) model (Wigneron et al., 2007), which uses the Tau-Omega  $(\tau - \omega)$  radiative 83 transfer equation to simulate surface microwave emission (Cui et al., 2015; Mo et al., 1982). SMOS-IC 84 is the latest algorithm in this series, which does not rely on auxiliary vegetation information as initial 85 inputs but uses the annual average of previously retrieved vegetation  $\tau$  during the retrieval process 86 (Li et al., 2022a). The latest release of SMOS-IC v2 further improves upon this by incorporating a 87 first-order modeling approach (2-Stream) instead of the zero-order  $\tau - \omega$  model (Li et al., 2020). 88

The Soil Moisture Active Passive (SMAP) mission is to monitor the dynamics of soil moisture and vegetation moisture content globally, launched by the National Aeronautics and Space Administration (NASA) in 2015 (Entekhabi et al., 2010; Le Vine et al., 2010). SMAP carries an active microwave radiometer that emits microwave signals and then uses the reflection and scattering data from the signals to calculate parameters such as SM and VWC. Currently, SMAP retrieval algorithms are primarily categorized into single-channel algorithms (SCA) (Jackson, 1993) and dual-channel algorithms (DCA) (Njoku et al., 2003) based on polarization. In contrast, DCA utilizes both H and V polarization channels and employs a nonlinear least squares optimization process to simultaneously retrieve SM and
L-VOD (Crow et al., 2005; O'Neill et al., 2018). Due to the correlated brightness temperature observations in dual-polarization channels, which cannot independently retrieve two unknowns, Koning et al. (Konings et al., 2016, 2017) proposed the Multi-Temporal Dual Channel Algorithm (MT-DCA) to
enhance the robustness of retrieval.

To sum up, the L-VOD retrieval algorithms for both SMOS and SMAP have reached a relatively 101 mature stage. Both sensors operate in fully polarised mode and have demonstrated a strong capability 102 in globally monitoring surface soil and vegetation characteristics. However, due to limitations such 103 as satellite scanning gaps and retrieval methods, the daily data provided by the two satellites are 104 spatially incomplete. This data missing phenomenon affects the seamless monitoring of VWC, above-105 ground biomass (AGB), etc. The seamless daily L-VOD data enhances the precision and timeliness of 106 vegetation change monitoring, enabling the capture of short-term environmental changes and sudden 107 events (e.g., extreme weather and natural disasters) impacts on vegetation. Currently, most applica-108 tions of VOD use multi-temporal data averaging. Incomplete VOD products are typically averaged 109 on monthly, quarterly, and annual scales to generate global coverage products (Olivares-Cabello et al., 110 2022; Wild et al., 2022). The drawbacks of the multi-temporal data averaging method are evident. It 111 compromises high temporal resolution, reducing the data utilisation. Additionally, the unique spatial 112 distribution of daily data is overlooked, leading to the loss of dense time-series variation information. 113 In other words, averaging VOD data over different time scales compromises the original information 114 in both spatial and temporal dimensions. 115

In order to overcome the missing data difficulties, recent studies have proposed reconstruction 116 methods of other products on a global or regional scale. Yang et al. (Yang and Wang, 2023) used the HCTSA method to extract the temporal features from surface SM time-series data, and then 118 reconstructed the data with the random forest model. Llamas et al. (Llamas et al., 2020) used 119 auxiliary data such as precipitation in combination with a multiple regression model to fill in the 120 blank portions of the CCI data. Zhang et al. (Zhang et al., 2021b) developed a novel spatiotemporal 121 partial convolutional neural network for AMSR2 soil moisture product gap-filling. Building on this 122 work, Zhang et al. (Zhang et al., 2022) proposed an integrated long short-term memory convolutional 123 neural network (LSTM-CNN), in which global daily precipitation datasets were fused into the proposed 124 reconstruction model to further improve gap-filling in daily soil moisture products. So far, there are 125 few works for L-VOD reconstruction on both global and daily scales. 126

In addition, SMOS satellite products are limited by coarse spatial resolution (25 km), which 127 cannot capture fine-scale phenological changes in surface vegetation. Although the SMAP satellite 128 improves spatial resolution, providing global L-VOD data at a 9 km resolution, it was launched in 129 2015 and therefore cannot provide historical data. To address the limitations of different sensors, 130 the recently released Vegetation Optical Depth Climate Archive (VODCA) version 2 (Zotta et al., 131 2024) combines VOD data from multiple sensors (SSM/I, TMI, AMSR-E, WindSat, and AMSR2) 132 to generate a long-term VOD product. Compared to the version 1 (Myneni et al., 2015), the main 133 improvement is the addition of L-band products (VODCA L) based on the SMOS and SMAP missions, 134 which are theoretically more sensitive to the entire canopy (including branches and trunks). However, 135 over extended periods such as 2010-2021, the spatial resolution of the existing L-VOD data remains 136 limited to 25 km. Currently, there are few studies that perform spatiotemporal fusion of the L-VOD 137 products from the two satellites to compensate for their spatiotemporal limitations. 138

In summary, current VOD products from different sources suffer from data gaps and coarse resolution of historical data. Hence the need to integrate multi-temporal and multi-source L-VOD products. Enhancing VOD quality by incorporating auxiliary data introduces more uncertainty. Independent retrieval of VOD products from microwave observations would be a more effective way to improve data quality. From these perspectives, our study begins with the reconstruction of missing data. Subsequently, a spatiotemporal fusion model is developed to generate seamless, long-term, 9-km global daily L-VOD products. The main contributions are below.

Based on the three-dimensionality (2-D spatial + time) spatiotemporal dataset, we reconstruct
 the missing parts of SMOS L-VOD data from 1 January 2010 to 31 December 2017 and SMAP L-VOD
 data from 1 April 2015 to 31 July 2021, filling a gap in the research field regarding global daily L-VOD
 products reconstruction.

2. A spatiotemporal fusion model based on the non-local filtering approach to generate a longterm 9-km L-VOD dataset. The fusion product is validated temporally and spatially, and numerically compared with the original 9-km data during the overlapping period. Based on the availability of existing data, we ultimately obtain a global daily seamless L-VOD dataset with the spatial resolution of 9 km for the period from 1 January 2010 to 31 July 2021.

3. The gap-filling accuracy is assessed using time series validation and simulated missing region validation. For the fusion products, temporal and spatial verification strategies are employed and numerical comparisons are made with the original 9-km data from the overlap period. Evaluation indexes demonstrate that the global daily seamless L-VOD dataset shows high accuracy, reliability, and robustness.

The structure of this remaining paper as follows. Section 2 describes the L-VOD data and auxiliary data used in this study. Section 3 introduces the methods for gap filling and spatiotemporal fusion, as well as the experimental setup and accuracy validation metrics. Section 4 presents the experimental results and relevant validation results. Finally, Section 5 provides the conclusions of this study and suggestions for future work.

# <sup>165</sup> 2 Data description

### <sup>166</sup> 2.1 L-VOD data

SMOS IC L-VOD dataset is published by the European Space Agency (ESA) and has a satellite 167 revisit period of 8 days, a spatial resolution of 25 km, and a global spatial coverage. This study uses 168 the latest improved version 2 of L-VOD data for the period from 1 January 2010 to 31 December 2017, 169 which does not require the use of the optical vegetation index as an auxiliary data to drive the model, 170 enhancing the independence and stability of the product. This data is derived from https://ib.remote-171 sensing.inrae.fr/index.php/smos-ic-v2-product-documentation/ (Wigneron et al., 2021). Due to the 172 long-term advantage of SMOS L-VOD data, it is used as the low spatial resolution data for both 173 174 the reference and target periods in the spatiotemporal fusion experiments. This data participates in constructing the baseline data and assists in generating 9-km L-VOD data for the target moments. 175

SMAP MT-DCA L-VOD dataset covers the global surface with a satellite revisit period of 3 days 176 and a spatial resolution of 9 km. This study uses the latest SMAP MT-DCA version 5 L-VOD data 177 released by Feldman et al. (Feldman and Entekhabi, 2019), which updates the data from 1 April 178 2015 to 31 July 2021. This data is derived from https://doi.org/10.5281/zenodo.5619583 (Feldman 179 et al., 2021). The MT-DCA algorithm combines microwave radiometer data from the SMAP satellite 180 and vegetation index data from MODIS, while also considering the temporal autocorrelation of VOD. 181 Similar to the SMOS IC algorithm, MT-DCA does not require optical auxiliary data to provide initial 182 VOD values due to its consideration of VOD's temporal autocorrelation. SMAP L-VOD data has the 183 advantage of high spatial resolution, which is used in this study as the high-resolution baseline data in 184 the spatiotemporal fusion model to provide fine spatial detail information for the VOD fusion product. 185

<sup>186</sup> A specific description of the L-VOD data is shown in Table 1.

Table 1. Description of L-VOD data used in this study

| Product | Source      | Version | Temporal and spatial resolution | Period                |
|---------|-------------|---------|---------------------------------|-----------------------|
| L-VOD   | SMOS IC     | V2      | 25  km/daily                    | 2010.1.1-2017.12.31   |
| L-VOD   | SMAP MT-DCA | V5      | $9 \mathrm{~km/daily}$          | 2015.4.1- $2021.7.31$ |

#### <sup>187</sup> 2.2 Auxiliary data

To carry out the relevant analysis more comprehensively and accurately, we use two important auxiliary datasets, namely land cover types data and Normalized Difference Vegetation Index(NDVI) data.

This study selected pixel points under different land cover types for accuracy validation. The data is based on the MODIS MCD12C1 V061 (Friedl and Sulla-Menashe, 2022), which provides global land

cover types at annual intervals with a time span from 2001 to 2022 and a spatial resolution of  $0.05^{\circ}$ 193 (approximately 5.6 km). This dataset uses multiple classification schemes, including International 194 Geosphere-Biosphere Programme(IGBP), University of Maryland(UMD), and Leaf Area Index(LAI) 195 (Chen and Black, 1992; Hansen et al., 2000; Loveland et al., 1999). In this study, land cover data 196 for 2017 and 2018 are used. The data is accessed and processed through the Google Earth Engine 197 platform. 198 In this study, we choose long-term NDVI data to further evaluate the final product VOD\_st. The 199 data is based on the MODIS MYD13C1 V061 (Didan, 2021), which has a spatial resolution of 0.05° 200 (approximately 5.6 km) and is synthesized over 16 days. This product provides a Vegetation Index 201 (VI) value for each pixel, namely the Enhanced Vegetation Index (EVI) and the NDVI. We use the 202 NDVI data from 2010 to 2021, which maintains continuity with the existing National Oceanic and 203 Atmospheric Administration-Advanced Very High Resolution Radiometer (NOAA-AVHRR) derived 204

205 NDVI.

Considering the availability of the dataset, the study period for this research is from 1 January 2010 to 31 July 2021. For convenience, the original SMOS IC L-VOD product is referred to as VOD\_smos, the original SMAP MT-DCA L-VOD product as VOD\_smap, the gap filling products as VOD\_resmos and VOD\_resmap, respectively, and the spatiotemporal fusion product as VOD\_st.

# 210 3 Methodology

#### <sup>211</sup> 3.1 Data preprocessing

For the selected VOD\_smos and VOD\_smap datasets, preprocessing steps such as reprojection, 212 anomaly handling, and resampling are required. Due to differences in geographic coverage and pro-213 jection methods between SMOS and SMAP data products, reprojection is necessary. Additionally, 214 considering that VOD typically ranges from 0 to 1.5, with higher values often observed in densely 215 vegetated tropical regions, reaching up to approximately 1.2, there are occasional outliers exceeding 216 1.5 in specific areas like the Amazon and Congo river basins, accounting for approximately 1% of 217 the total (Fernandez-Moran et al., 2017; Li et al., 2022a). To minimize the potential accumulation 218 of uncertainty in subsequent experiments caused by abnormal values, these data need to be removed. 219 Furthermore, some regions may have negative VOD values due to unreliable retrieval caused by sen-220 sor limitations or land types such as permafrost or deserts. VOD values less than zero cannot be 221 explained by physical properties. Following the guidelines from Wigneron et al. for the SMOS IC 222 L-VOD data (https://ib.remote-sensing.inrae.fr/index.php/smos-ic-v2-product-documentation/), neg-223 ative VOD values will be set to zero in this study to ensure result accuracy. Lastly, the low-resolution 224 product VOD\_smos will be preliminarily resampled to 9 km using nearest neighbor interpolation 225 to maintain consistency in spatial resolution across all datasets. Our data utilize a global grid of 226  $2000 \times 4000$  cells. 227

We consider that VOD has continuity over long temporal sequences but faces a significant proportion of spatial data gaps. Moreover, in the spatiotemporal fusion model, higher spatial coverage of input data, represented by a larger effective number N, leads to better spatiotemporal fusion effects. Therefore, our study proposes initially using a penalized least square regression based on three-dimensional discrete cosine transform (DCT-PLS) method to leverage spatiotemporal variation information for repairing L-VOD data from SMOS and SMAP satellites. Subsequently, seamless data will be input into a non-local filter based spatiotemporal fusion model (STFM) model to reconstruct historical 9-km data, aiming to maximize error reduction and enhance product quality.

### <sup>236</sup> 3.2 Gap filling

Given the significant spatial data gaps in the VOD\_smos and VOD\_smap datasets, and considering that frequency domain signal distribution is more concentrated and contains more comprehensive information, the discrete cosine transform (DCT) is an effective algorithm for transforming signals into the frequency domain for computation (Wang et al., 2023). Additionally, penalized least square (PLS) regression is a thin-plate spline smoothing method suitable for one-dimensional arrays, which aims to balance data fidelity and the roughness of the mean function. Garcia (Garcia, 2010) has demonstrated that DCT achieves PLS regression by expressing data as a sum of cosine functions oscillating at different frequencies. Due to the multidimensional characteristics of DCT, DCT-based
PLS regression can be directly extended to multidimensional datasets (Wang et al., 2012). For large
spatiotemporal datasets, utilizing spatiotemporal variation information to predict missing parts is
highly effective. Furthermore, VOD data shows significant temporal and spatial correlations, and
DCT can capture this spatiotemporal correlation well. Therefore, this study uses the three-dimensional
DCT-PLS method to fill the gaps in the global daily L-VOD data. The following section will briefly
introduce the principles of the DCT-PLS algorithm for data repair:

Let x represent the spatiotemporal dataset with missing values. The solution formula for the filled data matrix y is as follows:

$$F(y) = \left\| Q^{1/2} \cdot (y - x) \right\|^2 + \lambda \left\| \nabla^2 y \right\|$$
(1)

where  $\|\cdot\|$  denotes the Euclidean norm. Q is a binary matrix indicating the missing values in the original data, with the square root used for weight adjustment.  $\nabla^2$  is the Laplacian operator.  $\lambda$  is the smoothness factor, which measures the smoothness of the data y. The iterative solution for y can be transformed into the following formula:

$$y = \mathrm{DCT}^{-1}(G \cdot \mathrm{DCT}(Q \cdot (x - y) + y))$$
(2)

In this context, DCT is used to transform the data from the spatial domain to the frequency domain, where the data is then reconstructed. Finally, the inverse transform  $(DCT^{-1})$  is applied to convert the reconstructed results back from the frequency domain to the spatial domain. *G* is a three-dimensional filtering tensor:

$$G_{(k_1,k_2,k_3)} = \frac{1}{1 + \lambda (\sum_{m=1}^{3} (2 - \cos \frac{(k_m - 1)\pi}{N_m}))^2}$$
(3)

where  $k_m$  represents the k-th element in the m-th dimension (where m = 1, 2, 3), and  $N_m$  denotes the size of the data in the m-th dimension of the matrix x.

In DCT-PLS modeling, the selection of the smoothing parameter  $\lambda$  is crucial. A higher value of the smoothing parameter will result in the loss of high-frequency components. To effectively fill in the data gaps,  $\lambda$  should be as close to zero as possible to minimize the smoothing effect. By calculating the normalized error between the original and reconstructed values, it can be determined whether the model accurately captures the characteristics of the data. Thus, the smoothing parameter  $\lambda$  can be adjusted based on the error evaluation results to optimize model performance. The error  $\epsilon$  is defined as follows:

$$\epsilon = \frac{\|Q^{1/2} \cdot (y - x)\|}{\|Q^{1/2} \cdot x\|} \tag{4}$$

#### <sup>270</sup> 3.3 Spatiotemporal fusion

Spatiotemporal fusion of remote sensing data is the process of integrating multi-source remote 271 sensing data into products that have spatiotemporal consistency and higher accuracy. Among these 272 methods, both transformation-based and pixel-based reconstruction methods are commonly used ap-273 proaches (Belgiu and Stein, 2019; Zhu et al., 2018). Transformation-based methods include techniques 274 such as Fourier transform and wavelet transform (Fanelli et al., 2001; Gharbia et al., 2014). These 275 methods fuse data by combining transform coefficients from different sources, offering simplicity and 276 ease of implementation. However, they often suffer from lower accuracy and are prone to introduc-277 ing noticeable artifacts in the fusion images. On the other hand, pixel-based reconstruction methods 278 involve weighted averaging or other operations on pixel values from different source data to achieve 279 fusion. This approach has become the mainstream method in current spatiotemporal fusion research 280 due to its ability to preserve spatial details and improve overall accuracy. Within these methods, 281 the spatial and temporal adaptive reflectance fusion model (STARFM) has been widely applied (Gao 282 al., 2006). An improved approach to the STARFM model is used in this study. et 283

This study aims to extend the SMAP 9-km VOD by developing a non-local filter based spatiotemporal fusion model (STFM) (Cheng et al., 2017). This model employs the transformation relationships <sup>286</sup> between high-resolution spatial and low-resolution temporal data over different time periods to effectively utilize the high spatiotemporal correlation in remote sensing image sequences for predicting
<sup>287</sup> high spatial resolution data at the target time. For convenience, in this study, we refer to images
<sup>289</sup> with high spatial resolution and low temporal resolution as high-resolution images, and conversely, as

low-resolution images, based on spatial resolution as the criterion.

As mentioned above, this experiment performs spatiotemporal fusion on the reconstructed data VOD\_resmos and VOD\_resmap to obtain the VOD\_st product. Assuming that the changes in VOD are linear over a short period, the relationship between the data at different times  $t_k$  and  $t_0$  within a pixel can be expressed as follows:

$$VOD\_resmos(x, y, t_k) = a(x, y, \Delta t) \cdot VOD\_resmos(x, y, t_0) + b(x, y, \Delta t)$$
(5)

where (x, y) denotes a given pixel location in the low-resolution data,  $\Delta t = t_k - t_0$ , and a and b are the coefficients of the linear regression model describing the change in VOD\_resmos between the two time points.

We assume that the high- and low-resolution data obtained by different sensors in the same spectral band exhibit similar temporal variations. Thus, the linear relationship between low-resolution remote sensing images, as shown in Eq.(5), also applies to high-resolution remote sensing images. The high-resolution data at time  $t_k$  can be calculated as:

$$VOD_st(x, y, t_k) = a(x, y, \Delta t) \cdot VOD_resmap(x, y, t_0) + b(x, y, \Delta t)$$
(6)

It should be noted that the regression coefficients are derived locally and may vary with location. Hence, they cannot be applied globally. Additionally, the condition of the surface cover might undergo significant and complex changes during the prediction period. Therefore, the STFM algorithm incorporates a new non-local filtering method to minimize the impact of these factors on the fusion outcome.

The non-local filtering method seeks to make full use of the highly redundant information within the image, thus contributing to the estimation of the target pixel (Buades et al., 2005a,b; Gilboa and Osher, 2009; Su et al., 2012). Within the search window  $\Omega$ , the similarity between neighboring pixels and the central pixel will influence the determination of the weights. The weight calculation method is as follows:

$$W(x_i, y_i) = \frac{1}{C(x, y)} \exp\left\{-\frac{G \cdot \|\text{VOD\_resmos}(P(x_i, y_i)) - \text{VOD\_resmos}(P(x, y))\|^2}{h^2}\right\}$$
(7)

Where C(x, y) is the normalization factor, G is the Gaussian kernel, and h is the filtering parameter. The term  $(x_i, y_i) \in \Omega$  represents the coordinates of neighboring pixels within the search window, and  $P_{(x_i, y_i)}$  is the non-local similarity patch centered at  $(x_i, y_i)$ . Once the similar pixels are determined globally, their information is used for estimating the target pixel through weighted averaging. The final spatiotemporal fusion prediction model can be expressed as follows:

$$\text{VOD\_st}(x_i, y_i, t_k) = \sum_{i=1}^{n} W(x_i, y_i, t_0) \times [a(x_i, y_i, \Delta t) \times \text{VOD\_resmap}(x_i, y_i, t_0) + b(x_i, y_i, \Delta t)]$$
(8)

 $_{317}$  Where *n* represents the number of similar pixels globally.

Since VOD\_smos data is available from 1 January 2010 to the present, while VOD\_smap data 318 319 covers the period from 1 April 2015 to 31 July 2021. To fill the temporal blank in high spatial resolution L-VOD products before the launch of the SMAP satellite, we use 1 April 2015, the initial 320 date provided by the VOD\_smap product, as the time node. The time range to be predicted by the 321 VOD<sub>st</sub> product is defined as the T1 period, spanning from 1 January 2010 to 31 March 2015. To 322 construct the baseline data required for the spatiotemporal fusion model and considering the temporal 323 correlation, we extend one year beyond the fusion input period, defining the T2 period from 1 April 324 2015 to 1 April 2016. To validate the quality of the fusion product VOD\_st, we define the remaining 325 period from 2 April 2016 to 31 December 2017 as the T3 period. For specific details, refer to Fig. 1. 326 Fig. 2 illustrates that the spatiotemporal fusion model requires paired high- and low-resolution

Fig. 2 illustrates that the spatiotemporal fusion model requires paired high- and low-resolution data to construct the baseline data. To achieve a more temporally correlated fusion product, we use

Fig. 1. Spatiotemporal fusion experiment time segment division explanation.

monthly averaged VOD\_resmos and VOD\_resmap from April 2015 to April 2016 to generate baseline
data, which is a key step in learning the transformation relationships between high - resolution and low
- resolution data across different periods. Subsequent experiments utilize this baseline data, inputting
daily low-resolution VOD\_resmos data for each corresponding month to obtain daily high-resolution
spatiotemporal fusion product VOD\_st.

Fig. 2. Spatiotemporal fusion Process.

In summary, this study first utilizes the DCT-PLS method to fill gaps in the original missing data, obtaining the reconstructed products, the VOD\_resmos and VOD\_resmap. Subsequently, the reconstructed global seamless daily datas are input into the spatiotemporal fusion model STFM, generating the 9-km VOD\_st product for unreleased periods of the SMAP satellite. The main experimental process is illustrated in Fig. 3. The accuracy validation part is detailed in Section 4.

#### 339 3.4 Experimental Setup

In this study, a three-dimensional dataset (2D spatial + time) is constructed with a monthly time series length. The DCT-PLS method is an iterative algorithm designed to fill missing values in multi-dimensional data. In this experiment, the number of iterations is set to 100, with the initial prediction of the original data performed using the nearest neighbor interpolation method. The smoothing parameter ( $\lambda$ ) follows a logarithmic sequence from  $10^{-3}$  to  $10^{-6}$ . During the imputation process, the algorithm gradually reduces the smoothing parameter to achieve a transition from coarse to fine imputation.

The STFM algorithm processes data in batches, using the high- and low-resolution monthly average baseline data constructed for the T2 period, along with the daily low-resolution data for the corresponding month at the target time. After multiple adjustments, the optimal combination of parameters for the L-VOD data is determined. Table2 describes the meaning and specific values of these parameters.

The quantitative evaluation metrics used in the experimental section of this study include five