# Peer review of "A global daily seamless 9-km Vegetation Optical Depth (VOD) product from 2010 to 2021"

_Earth System Science Data, 2024_

## Referee Comment (RC1)

General Comments:

This study introduces a novel approach to reconstructing and merging global daily L-band Vegetation Optical Depth (L-VOD) data at a 9-km resolution from 2010 to 2021. The authors integrate datasets from the SMOS and SMAP satellites. They use a three-dimensional discrete cosine transform-based penalized least square regression (DCT-PLS) method to fill gaps in the data, followed by a spatiotemporal fusion model (STFM) based on non-local filtering to enhance the spatial and temporal continuity of the dataset. This material is important for long-term canopy and vegetation change studies. The manuscript is well-prepared, and the VOD data are valuable and should be published in *ESSD*. However, I have some questions about the data and their seasonal pattern, hope the authors can clarify them and confirm the data quality.

Specific Comments:

In Fig. 6, the description indicates that the rectangles emphasize some extreme value "reconstruction" results, but why does the first panel with a 30% missing proportion have a rectangle in the "Original" part line (magenta color)?

For Fig. 6 and Fig. 7, the colors for "Reconstructed" and "Original" are too similar. Please consider using more distinct colors, such as red and blue or black and blue. Additionally, the y-axis uses different scales in these figures. I suggest using the same y-axis scale range to better compare the relative size differences.

In Fig. 10, the x- and y-axes represent VOD. What do the colors and the color bar indicate? Please clarify their meaning.

In Fig. 18, VOD_st has more blank data compared to VOD_resmap. Can you provide a clearer explanation of why VOD_st is considered close to VOD_resmap (line 490)?

In Fig. 19, the red boxes and zoomed-in plots show that the reconstructed data appear smoother and free of striping. However, it is difficult to conclude that the reconstructed data are necessarily better. Why was the Black Sea region chosen as an example instead of another region with a higher VOD signal?

Data part:

1. Each data file includes only one time step and has a file size of 3.7 MB. I suggest merging them by year into a single NetCDF file.
2. I selected VOD data from two days: summer (2021-07-07, left panel) and winter (2021-01-07, right panel). In the Eastern US and many other regions, the seasonal pattern appears reasonable, with higher VOD in summer and lower VOD in winter in the Northern Hemisphere. However, in Sweden (red rectangle) in Europe, the summer VOD (~0.6) is lower than the winter VOD (~0.8). Additionally, the black boxes indicate that the western US, Alaska and western Canada have higher VOD in winter (almost as high as Amazon rainforest, VOD >1.0). Please confirm whether this seasonal pattern is reasonable in those regions.

3. There are large areas of missing data in Russia, China, and Japan during winter, but a specific spot in Russia appears red. Please confirm whether this pattern is reasonable.

---

## Author Comment (AC1)

**Response to Reviewer Comments**

**A global daily seamless 9-km Vegetation Optical Depth (VOD) product from 2010 to 2021**

Die Hu [a], Yuan Wang [b], Han Jing [a], Linwei Yue [c], Qiang Zhang [d], Lei Fan [e] [*], Qiangqiang Yuan [a, f, g, †], Huanfeng Shen [h], and Liangpei Zhang [i]

[a] School of Geodesy and Geomatics, Wuhan University, Wuhan, China.

[b] School of Atmospheric Physics, Nanjing University of Information Science and Technology, Nanjing, China.

[c] School of Geography and Information Engineering, China University of Geosciences, Wuhan, China.

[d] Center of Hyperspectral Imaging in Remote Sensing (CHIRS), Information Science and Technology College, Dalian Maritime University, Dalian, China.

[e] Chongqing Jinfo Mountain Karst Ecosystem National Observation and Research Station, School of Geographical Sciences, Southwest University, Chongqing, China.

[f] Key Laboratory of Geospace Environment and Geodesy, Ministry of Education, Wuhan University, Wuhan, China.

[g] Key Laboratory of PolarEnvironment Monitoring and Public Governance, Ministry of Education, Wuhan University, Wuhan, China.

[h] School of Resource and Environmental Science, Wuhan University, Wuhan, China.

[i] State Key Laboratory of Information Engineering in Surveying, Mapping, and Remote Sensing, Wuhan University, Wuhan, China.

[*] Corresponding authors. E-mail address: leifan33@swu.edu.cn

[†] Corresponding authors. E-mail address: qqyuan@sgg.whu.edu.cn

**Response to Comments of Reviewer #1:**

**General comments:** This study introduces a novel approach to reconstructing and merging global daily L-band Vegetation Optical Depth (L-VOD) data at a 9-km resolution from 2010 to 2021. The authors integrate datasets from the SMOS and SMAP satellites. They use a three-dimensional discrete cosine transform-based penalized least square regression (DCT-PLS) method to fill gaps in the data, followed by a spatiotemporal fusion model (STFM) based on non-local filtering to enhance the spatial and temporal continuity of the dataset. This material is important for long-term canopy and vegetation change studies. The manuscript is well-prepared, and the VOD data are valuable and should be published in ESSD. However, I have some questions about the data and their seasonal pattern, hope the authors can clarify them and confirm the data quality.

**Response:** We sincerely thank the reviewer for his/her positive comments. According to the concerns, we have tried our best to improve the manuscript. The point-by-point responses are listed below. Thanks for your time.

**Response to Specific Comments:**

**Q: 1) In Fig. 6, the description indicates that the rectangles emphasize some extreme value "reconstruction" results, but why does the first panel with a 30% missing proportion have a rectangle in the "Original" part line (magenta color)?**

**Response:** This is a good question. In Figure 6, the red line represents the original values and is overlaid on the blue line representing the reconstructed values. In other words, the original values are missing, while the reconstructed values are continuous. The results show that the DCT - PLS model does not alter the original pixel values themselves. It preserves the original characteristics of the data and maintains the continuity of the reconstructed results. In the first panel with a 30% missing proportion, the rectangle is marked on the original values, indicating that the reconstruction process can not only maintain the original non - missing parts but also capture extreme values well. Similar studies [1] also mark extreme values on the original values.

*References:*

[1] Wang, G., Garcia, D., Liu, Y., De Jeu, R., and Dolman, A. J.: A three-dimensional gap filling method for large geophysical datasets: Application to global satellite soil moisture observations, Environmental Modelling & Software, 30, 139–142, 2012.

**Q: 2)** **For Fig. 6 and Fig. 7, the colors for "Reconstructed" and "Original" are too similar. Please consider using more distinct colors, such as red and blue or black and blue. Additionally, the yaxis uses different scales in these figures. I suggest using the same y-axis scale range to better compare the relative size differences.**

**Response:** Thanks for the reviewer's suggestion regarding the detail of our mapping. According to the advice, we have changed the colors of the reconstructed values and the original values to distinct blue and red. The extreme values are marked with black boxes.

[Figure]

**Fig. 1.** Results of temporal variation in selected pixel at different missing data ratios in 2018, with red representing original values, blue representing model reconstructed values, and rectangles emphasizing some extreme value reconstruction results.

In the comparison of the temporal variation results of different land cover types, to ensure consistency, we select pixels with a data - missing proportion of 52% throughout the year for analysis.

[Figure]

**Fig. 2.** The red dots in the figure indicate the pixel points selected to characterise the temporal variation of L-VOD under different vegetation conditions. Four different surface types are selected here, namely (a) scrub, (b) forest, (c) cropland, and (d) grassland; (1)-(4) represent the time-series variation maps of the corresponding pixels under the above surface types, respectively.

**Q: 3)** In Fig. 10, the x- and y-axes represent VOD. What do the colors and the color bar indicate? Please clarify their meaning.

**Response:** Thanks for the comment. The colors and the color bar indicate the density of data points in the scatter plot. The color bar indicates the range of density, with the colors transitioning from blue to red, where blue represents lower density and red indicates higher density. This helps visualize the distribution of the original and reconstructed VOD values, with more frequent data points represented by warmer colors (red) and less frequent points represented by cooler colors (blue).

**Q: 4)** In Fig. 18, VOD_st has more blank data compared to VOD_resmap. Can you provide a clearer explanation of why VOD_st is considered close to VOD_resmap

**(line 490)?**

**Response:** Thanks for the comment. During the spatiotemporal fusion process, VOD_st learns the temporal characteristics of VOD_resmos (from 2010 to 2015) and the spatial characteristics of VOD_resmap (with a 9-km spatial resolution). During the period from 2010 to 2015, only the SMOS satellite provides L-band VOD products. Therefore, the spatial coverage of VOD_st from 2010 to 2015 is completely dependent on VOD_resmos. In comparison, the spatial coverage of the SMOS satellite products is not as extensive as that of the SMAP satellite. In our paper, VOD_st is considered closer to VOD_resmap (line 490) in terms of numerical accuracy. Specifically, it is closer to the high-resolution product in terms of numerical performance of the valid data (9 km) rather than in terms of spatial coverage (VOD_resmos). This numerical proximity further demonstrates that the fusion product (VOD_st) has a certain degree of reliability in reflecting relevant features.

**Q: 5) In Fig. 19, the red boxes and zoomed-in plots show that the reconstructed data appear smoother and free of striping. However, it is difficult to conclude that the reconstructed data are necessarily better. Why was the Black Sea region chosen as an example instead of another region with a higher VOD signal?**

**Response:** Thanks for the comment. We select the Black Sea region as an example due to its representative ecosystem, which primarily consists of grasslands and croplands. Moreover, the proportion of missing data in this area is moderate, mostly ranging from 40% to 50%. Since the percentage of missing data is not very high, the data distribution in the region is relatively uniform. The data characteristics are generally consistent, reducing the impact of extreme values or unusual data clusters. Therefore, the difference between the monthly averages before and after reconstruction is not significant.

In order to better compare the results before and after the reconstruction, we have re-selected a relatively more representative area, the Kalimantan Island (5° S - 8° N, 108° E - 120° E). The VOD signals on Kalimantan Island are higher, and the missing -data proportion mainly ranges from 50% to 80%, which can better reflect the

reconstruction ability. Kalimantan Island is characterized by its large - area and diverse - type tropical rainforests. The dynamic changes in vegetation are significantly affected by human activities. Located in the tropical climate zone, it has complex climatic conditions, abundant precipitation, and extreme weather events that can impact vegetation. With diverse landforms and a special geographical location, as well as social and economic activities such as agricultural development and eco - tourism, this island becomes a typical area for testing the effectiveness and reliability of the reconstruction method in complex environments.

In the selected local area, the original data presents blocky patterns. There are significant differences in VOD values between different patches, and the edges are rather abrupt. Meanwhile, there may be some noises or outliers in the original data, resulting in a non - smooth spatial distribution of the data. The reconstructed data shows a smoother spatial transition. It indicates that the reconstruction algorithm not only fills in the missing data values but also processes the noises and outliers in the original data to a certain extent. It is important to note that VOD products exhibit insignificant variations at the daily scale. So the difference between the monthly average data before and after the reconstruction is not significant.

[Figure]

**Fig. 3.** Original (top) and reconstructed (bottom) results for July 2015 SMOS VOD monthly average. The red boxes highlight local areas.

**Response to Data part Comments:**

**Q: 1) Each data file includes only one time step and has a file size of 3.7 MB. I suggest merging them by year into a single NetCDF file.**

**Response:** We sincerely thank the reviewer for the suggestion. We have followed this suggestion and merged the daily data within each year into one NetCDF file. The variable names are named as VOD_xxxxyydd, where xxxx represents the year, yy represents the month, and dd represents the day. The longitude variable is named "lon" with a dimension of 4000×1, and the latitude variable is named "lat" with a dimension of 2000×1. It should be noted that these NetCDF files are saved using the netCDF4 library in Python, with the dimension order being (lat, lon). When reading these NetCDF files in MATLAB, the default data dimension order is (lon, lat). Therefore, it is necessary to transpose the variables to match the correct dimension order.

**Q: 2) I selected VOD data from two days: summer (2021-07-07, left panel) and winter (2021-01-07, right panel). In the Eastern US and many other regions, the seasonal pattern appears reasonable, with higher VOD in summer and lower VOD in winter in the Northern Hemisphere. However, in Sweden (red rectangle) in Europe, the summer VOD (~0.6) is lower than the winter VOD (~0.8). Additionally, the black boxes indicate that the western US, Alaska and western Canada have higher VOD in winter (almost as high as Amazon rainforest, VOD >1.0). Please confirm whether this seasonal pattern is reasonable in those regions.**

**Response:** Thank you for your careful review and the valuable comment. We appreciate your attention to the seemingly unusual seasonal patterns of VOD in specific regions. Regarding the situation in Sweden where the summer VOD (~0.6) is lower than the winter VOD (~0.8), this phenomenon can be reasonably explained by several factors. In Sweden, during the winter, although the vegetation may be less physiologically active, the presence of snow cover can have a significant impact on the VOD measurement. Snow has unique electromagnetic properties, and its high dielectric constant can lead to increased microwave backscattering, which in turn can elevate the VOD value [1]. In contrast, during the summer, although the vegetation is growing, the

relatively sparse forest cover, combined with possible effects of soil moisture and vegetation structure changes, might result in a lower VOD compared to the winter with snow cover.

As for the regions in the western US, Alaska, and western Canada where the winter VOD is higher (almost as high as in the Amazon rainforest, VOD >1.0), this can be attributed to the local environmental conditions. A large part of these areas is covered by coniferous forests. These forests are characterized by evergreen trees such as spruce, pine, and fir. The evergreen nature of these trees means that they retain their foliage throughout the winter [2]. Compared to deciduous trees that shed their leaves in winter, the continuous presence of foliage in coniferous forests leads to a relatively stable and high VOD [3]. In addition, the density of coniferous forests in these regions is relatively high in many areas. This high - density vegetation further contributes to the elevated VOD values, making them comparable to those in the Amazon rainforest in terms of magnitude.

In conclusion, based on the local environmental characteristics and the influence mechanisms of various factors on VOD, the observed seasonal patterns in these regions are reasonable and consistent with our understanding of the interaction between the environment and microwave remote sensing signals.

*References:*

[1] Mätzler C. Applications of the interaction of microwaves with the natural snow cover[J]. Remote sensing reviews, 1987, 2(2): 259-387.

[2] Tian F, Wigneron J P, Ciais P, et al. Coupling of ecosystem-scale plant water storage and leaf phenology observed by satellite[J]. Nature ecology & evolution, 2018, 2(9): 1428-1435.

[3] Jones H G, Vaughan R A. Remote sensing of vegetation: principles, techniques, and applications[M]. OUP Oxford, 2010.

**Q: 3) There are large areas of missing data in Russia, China, and Japan during winter, but a specific spot in Russia appears red. Please confirm whether this**

**pattern is reasonable.**

**Response:** We sincerely thank the reviewer for pointing out this interesting observation. In the final L-VOD product, the data from April 1, 2015, to July 31, 2021, is seamless data obtained through gap - filling model. The VOD product from January 1, 2010, to March 31, 2015, is further processed using a spatiotemporal fusion algorithm based on the seamless data. Both of these datasets maintain spatial consistency with the original satellite products.

As shown in Figure 4, which presents the monthly average of VOD_resmap in January 2021, we can observe that the SMAP satellite product has missing data during this period. Our reconstruction model constructs three - dimensional data on a monthly basis to learn spatiotemporal features. Since the input data (the original SMAP satellite data) has missing values, the reconstruction results also exhibit missing data. This is the reason for the large - scale missing data in Russia, China, and Japan during winter in our product.

The presence of the red spot in Russia can be attributed to the availability of original satellite data in that area. Unlike the surrounding regions in Russia, China, and Japan where there are large - scale data gaps during winter, this particular location has accessible original satellite data. The sensors are able to capture signals in this area, enabling reconstruction model to generate a valid VOD value.

In summary, these spatial patterns are reasonable given the characteristics and spatial consistency of the original data.

[Figure]

**Fig. 4.** This chart shows the monthly average of VOD_resmap in January 2021.

---

## Author Comment (AC2)

**Response to Reviewer Comments**

**A global daily seamless 9-km Vegetation Optical Depth (VOD) product from 2010 to 2021**

Die Hu [a], Yuan Wang [b], Han Jing [a], Linwei Yue [c], Qiang Zhang [d], Lei Fan [e] *, Qiangqiang Yuan [a, f, g, †], Huanfeng Shen [h], and Liangpei Zhang [i]

[a]  School of Geodesy and Geomatics, Wuhan University, Wuhan, China.

[b]  School of Atmospheric Physics, Nanjing University of Information Science and Technology, Nanjing, China.

[c]  School of Geography and Information Engineering, China University of Geosciences, Wuhan, China.

[d]  Center of Hyperspectral Imaging in Remote Sensing (CHIRS), Information Science and Technology College, Dalian Maritime University, Dalian, China.

[e]  Chongqing Jinfo Mountain Karst Ecosystem National Observation and Research Station, School of Geographical Sciences, Southwest University, Chongqing, China.

[f]  Key Laboratory of Geospace Environment and Geodesy, Ministry of Education, Wuhan University, Wuhan, China.

[g]  Key Laboratory of PolarEnvironment Monitoring and Public Governance, Ministry of Education, Wuhan University, Wuhan, China.

[h]  School of Resource and Environmental Science, Wuhan University, Wuhan, China.

[i]  State Key Laboratory of Information Engineering in Surveying, Mapping, and Remote Sensing, Wuhan University, Wuhan, China.

\* Corresponding authors. E-mail address: leifan33@swu.edu.cn

† Corresponding authors. E-mail address: qqyuan@sgg.whu.edu.cn

**Response to Comments of Reviewer #2:**

**Response to General comments:**

**Q: 1) A more detailed discussion of the results would strengthen the manuscript. In particular, further analysis of the performance of the new product across different land cover types would be beneficial. Additionally, evaluating VOD against vegetation-related parameters, such as aboveground biomass (AGB), NDVI, and LAI, would enhance clarity.**

**Response:** Thanks for the comment. Regarding the first point about a more detailed discussion of the results, especially the performance of the new product across different land cover types, we have already conducted relevant analyses. As shown in Figure 7 of our paper, we present the temporal variation results for four selected land cover types: forest, shrubland, cropland, and grassland. This analysis allows us to understand how the new product behaves differently under various land cover conditions, providing a solid basis for discussing its performance in different environments.

For the second point about evaluating VOD against vegetation - related parameters, we add a related experiment in which we compare VOD_st with NDVI. The monthly average comparison results are shown in the Figure 1. We can observe that the seasonal trends of VOD_st and NDVI are highly consistent, showing obvious periodic characteristics. During the summer months corresponding to the period of maximum vegetation growth and leaf production, the values of these parameters increase significantly, and they decline as the vegetation ages. This consistency indicates that VOD_st can effectively capture the changes in vegetation growth, similar to traditional optical - based indices like NDVI. Notably, VOD_st exhibits a slight lag in its seasonal changes compared to NDVI, but this lag is not due to the quality of VOD_st. Our findings are in line with previous studies by Lawrence et al. [1] and Xiaojun Li et al. [2], which also reported that VOD data has a slight lag when compared with optical vegetation indices.

The reasons for this lag are related to their distinct biophysical meanings. Firstly, NDVI is highly sensitive to rapid changes in leaf - level characteristics such as chlorophyll content and leaf area as it is based on the reflection and absorption of visible

and near - infrared light by the vegetation canopy. In contrast, VOD_st, relying on microwave - vegetation interactions, reflects more comprehensive and large - scale vegetation structural information and responds more to gradual changes in the overall vegetation structure over a longer time frame. Secondly, NDVI is mainly influenced by the optical properties of vegetation and is less directly affected by moisture in the short - term, while VOD_st is highly sensitive to changes in vegetation moisture content and the scattering and absorption properties of the medium. The time it takes for moisture - related changes to impact VOD_st compared to the relatively instantaneous optical changes captured by NDVI contributes to the lag. Thirdly, differences in temporal resolution and data processing between the two parameters can also lead to the non - alignment of their peaks and troughs.

Overall, this comparison between VOD_st and NDVI provides valuable insights into the relationship between microwave - based VOD and optical - based NDVI, helping to better understand the characteristics and performance of the VOD product.

[Figure]

**Fig. 1.** Long - term monthly average trend comparison between VOD and NDVI.

***References:***

[1] Lawrence H, Wigneron J, Richaume P, et al. Comparison between SMOS Vegetation Optical Depth products and MODIS vegetation indices over crop zones of the USA[J]. Remote Sensing of Environment. 2014, 140: 396-406.

[2] Li X, Wigneron J, Frappart F, et al. Global-scale assessment and inter-comparison of recently developed/reprocessed microwave satellite vegetation optical depth products[J]. Remote Sensing of Environment. 2021, 253: 112208.

**Q: 2)** **It may be helpful to consider measures to reduce the bias between SMOS VOD and SMAP VOD, as this discrepancy could impact the accuracy of the fused product. A more detailed analysis of this uncertainty would be valuable.**

**Response:** Thank you for highlighting the importance of uncertainty analysis for our fused product. First I would like to illustrate how bias between SMOS and SMAP VOD affect the results. SMOS and SMAP sensors have different observational capabilities, and the differences in instrumentation result in different ways of sensing and measuring VOD. In addition, the two have different VOD retrieval algorithms, which can also cause bias. The bias between SMOS and SMAP VOD products may introduce errors during the data fusion process, thereby affecting the accuracy and reliability of the fused product [1].

In the context of our study, we focus on the overall temporal and spatial trends of VOD rather than eliminating the bias between the two sensors' products. This is based on an assumption that within the same spectral band, high - resolution and low - resolution data obtained from different sensors have similar temporal changes.

We believe that these similar temporal variations can still provide valuable information for our research objectives. For instance, when analyzing the long - term trends of vegetation dynamics or the response of vegetation to environmental changes, the common temporal patterns in SMOS and SMAP VOD data can be used to draw meaningful conclusions. In addition, our study is more concerned with the general performance and usability of the fused product. We believe that the bias does not significantly distort the overall patterns and relationships.

We understand the importance of the bias issue and acknowledge that it may be necessary to further explore ways to mitigate bias in future studies for more accurate and refined results. However, in the scope of this current study, our approach based on the assumption of similar temporal variations is a valid strategy.

This bias can also lead to uncertainty in the final product. We can identify the following sources of uncertainty for the fused VOD product:

1. *The errors of original SMOS VOD and SMAP VOD products*. Our fused VOD product is generated based on the original SMOS VOD and SMAP VOD products.

These original datasets inherently contain errors, due to the satellite sensor's performance and the difference of VOD retrieval algorithms. These errors from the original data sources are propagated into our fused product, affecting its accuracy. In addition, we perform a gap-filling process on the original data, which also introduces uncertainty and increases the error in the final fused product.

2. *The meteorological factors.* Meteorological factors can affect vegetation phenology. Vegetation phenology plays a crucial role. For instance, rapid changes in vegetation growth stages, such as the sudden onset of leaf senescence or new growth, can cause significant variations in VOD values. If our fusion method does not fully account for these rapid changes, it can lead to inaccuracies in the fused product. Moreover, the presence of clouds and aerosols can interfere with the satellite measurements of VOD which can introduce uncertainties into the final product.

3. *The generalization limitations of the fusion model.* Our spatiotemporal fusion model is trained using a specific set of data. However, there are differences between the training data and the actual data used for testing and generating the final product. For example, the land cover types in the testing data might have different spatial distributions or compositions compared to those in the training data. Different land cover types have distinct VOD responses, and if the model is not well - generalized to these variations, it can lead to uncertainties in the fused product. Additionally, the temporal coverage of the training data might not fully capture all the possible seasonal and interannual variations in VOD, which can limit the model's ability to accurately fuse data in different scenarios and contribute to the overall uncertainty of the final product.

***References:***

[1] Li X, Wigneron J P, Frappart F, et al. The first global soil moisture and vegetation optical depth product retrieved from fused SMOS and SMAP L-band observations[J]. Remote Sensing of Environment, 2022, 282: 113272.

**Q: 3) The overall readability of the manuscript could be improved, particularly in**

**terms of phrasing, organization, and paragraph structure. The authors may wish to have the manuscript reviewed by a native English speaker to refine grammar, style, and syntax.**

**Response:** Thank you for your feedback regarding the language presentation in our manuscript. We sincerely appreciate your careful reading and constructive comments.

We completely understand your concerns about the English fluency and readability. Please allow us to explain that we have already undergone multiple rounds of language editing, including:

1. Professional proofreading by colleagues fluent in academic English;
2. Grammar checking using advanced language tools (Grammarly and Hemingway Editor);
3. Several iterations of meticulous self-editing to improve clarity.

While we acknowledge that perfecting academic language remains challenging for non-native speakers, we have made our best effort to ensure the technical content is presented with precision and clarity. The current version represents what we believe to be the optimal balance between scientific accuracy and linguistic quality given our capabilities.

However, we fully respect your expert opinion. Should the manuscript be accepted pending minor revisions, we would be happy to collaborate with professional editing services to make final language improvements at the production stage.

**Response to Specific Comments:**

**Q: 1) Page 4, line 191. Define "IGBP, UMD and LAI" before their first use.**

**Response:** Thank you for your careful review and the valuable comment. We have taken your suggestion into consideration and have made the necessary corrections. On page 4, line 191, before the first use of "IGBP", "UMD", and "LAI", we have added definitions. Detailed explanations are provided below:

-IGBP (International Geosphere-Biosphere Programme) [1] refers to a global research initiative that developed a widely used classification scheme for land cover types based on satellite data.

-UMD (University of Maryland) [2] refers to the land cover classification system developed by the University of Maryland, which is based on multi-temporal satellite data and has been widely applied in various environmental studies.

-LAI (Leaf Area Index) [3] is a key parameter in vegetation studies, representing the total leaf area per unit of ground area, which is important for understanding vegetation structure and function.

*References:*

[1] Loveland T R, Zhu Z, Ohlen D O, et al. An analysis of the IGBP global land-cover characterization process[J]. Photogrammetric engineering and remote sensing, 1999, 65: 1021-1032.

[2] Hansen M C, DeFries R S, Townshend J R G, et al. Global land cover classification at 1 km spatial resolution using a classification tree approach[J]. International journal of remote sensing, 2000, 21(6-7): 1331-1364.

[3] Chen J M, Black T A. Defining leaf area index for non‐flat leaves[J]. Plant, Cell & Environment, 1992, 15(4): 421-429.

**Q: 2) Page 5, line 220. Define "DCT-PLS" before its first use.**

**Response:** Thanks for the suggestion to improve our paper. In response to the reviewer's comment regarding "DCT-PLS" on page 5, line 220, we have added the definition before its initial use. Detailed explanations are provided below:

DCT-PLS stands for Discrete Cosine Transform - Partial Least Squares. Discrete Cosine Transform (DCT) is a mathematical transformation that converts a signal from the spatial domain to the frequency domain. It is often used for data compression and feature extraction as it can represent the data in terms of its frequency components.

Partial Least Squares (PLS) is a statistical method that is used for dimensionality reduction and regression modeling. In the context of our research, PLS is employed to establish a relationship between different variables in the VOD data to fill gaps. It combines the advantages of DCT in extracting relevant features from the data and PLS in finding the optimal relationship between variables, aiming to reconstruct missing or

incomplete data in a more accurate and efficient manner.

**Q: 3) Page 7, line 288. The bias between SMOS and SMAP products should be considered (e.g., 10.1016/j.rse.2022.113272). This addition is relevant and could significantly broaden the manuscript's appeal.**

Response: Thank you for your suggestion regarding the consideration of the bias between SMOS and SMAP products. This question has already been discussed in comment 2 in General comments. The addition you suggested has been added to the Discussion part in our study.

SMOS and SMAP sensors have different observational capabilities, and the differences in instrumentation result in different ways of sensing and measuring VOD. In addition, the two have different VOD retrieval algorithms, which can also cause bias. The bias between SMOS and SMAP VOD products may introduce errors during the data fusion process, thereby affecting the accuracy and reliability of the fused product [1].

In the context of our study, we focus on the overall temporal and spatial trends of VOD rather than eliminating the bias between the two sensors' products. This is based on an assumption that within the same spectral band, high - resolution and low - resolution data obtained from different sensors have similar temporal changes.

We believe that these similar temporal variations can still provide valuable information for our research objectives. For instance, when analyzing the long - term trends of vegetation dynamics or the response of vegetation to environmental changes, the common temporal patterns in SMOS and SMAP VOD data can be used to draw meaningful conclusions. In addition, our study is more concerned with the general performance and usability of the fused product. We believe that the bias does not significantly distort the overall patterns and relationships.

*References:*

[1] Li X, Wigneron J P, Frappart F, et al. The first global soil moisture and vegetation optical depth product retrieved from fused SMOS and SMAP L-band observations[J].

Remote Sensing of Environment, 2022, 282: 113272.

**Q: 4)** **Page 7, Line 313. Please consider adding a more detailed description of the spatiotemporal fusion experiment. This should include the relationship between the reconstructed SMOS VOD and reconstructed SMAP VOD, the division of time segments, and the relationship between the fusion product and the reconstructed SMOS VOD and SMAP VOD.**

**Response:** Thank you for your valuable comment regarding the need for a more detailed description of the spatiotemporal fusion experiment.

The reconstructed SMOS VOD and SMAP VOD play distinct yet complementary roles in our spatiotemporal fusion approach. On the one hand, the reconstruction results offer long - term SMOS VOD products from 2010 to 2015, filling the temporal gap before the SMAP mission's start. This data provides a continuous record of VOD trends over a relatively long period, allowing us to capture the seasonal and inter - annual variations in vegetation properties. On the other hand, the reconstructed SMAP VOD provides high - resolution data (9 km). This spatial information enables us to resolve local - scale details in vegetation distribution and structure.

We would like to emphasize that the time segment division in our spatiotemporal fusion experiment is clearly presented in Fig.1 of our paper (Figure 2 in this response). The division is based on the launch dates of the SMOS and SMAP satellites. The SMOS VOD data has been available since January 1, 2010, while the SMAP VOD data is accessible from April 1, 2015, to July 31, 2021. To fill the temporal blank in 9-km spatial resolution L-VOD products before the launch of the SMAP satellite, we select April 1, 2015, the initial date when the SMAP VOD products become available, as the time node. We define the prediction period of the fused product VOD_st as T1, which spans from January 1, 2010, to March 31, 2015. To construct the baseline data required for the spatiotemporal fusion model and consider the temporal correlation, we extend the fusion input period by one year. The T2 period is defined from April 1, 2015, to April 1, 2016. For the purpose of validating the quality of the fused product VOD_st, we define the remaining period from April 2, 2016, to December 31, 2017, as the T3

period. By comparing the fused product with the actual data during this period, we can effectively evaluate the performance and reliability of the spatiotemporal fusion method.

[Figure]

**Fig. 2.** Temporal division of spatiotemporal fusion experiment.

Both the reconstructed SMOS VOD and SMAP VOD serve as the input data for our spatiotemporal fusion model. They are used to construct the baseline data for the model, which is a key step in learning the transformation relationships between high - resolution and low - resolution data across different time periods. By analyzing the co - variations between the SMOS and SMAP VOD data at different scales and time intervals, the model can identify patterns that are characteristic of the relationship between the two datasets. This learned relationship is then applied to predict the high - resolution VOD_st at the target time. As shown in Fig.3, we input daily low-resolution VOD_resmos for each corresponding month into the model. Once the model learns from the SMOS and SMAP VOD data during the training phase, it is able to predict the daily high-resolution fusion product VOD_st. Thus, the fusion product VOD_st combines the spatial and temporal complementarities of the reconstructed SMOS VOD and SMAP VOD.

[Figure]

**Fig. 3.** Spatiotemporal fusion Process.

**Q: 5) Page 12, Fig. 7. The text in this figure is too small.**

**Response:** Thanks for the comment. In response to your comment, we have carefully adjusted the proportion of Fig. 7 within the manuscript. By enlarging the figure, we have ensured that the text within it is now in a more harmonious size relative to the overall graphic. This adjustment has been made to optimize the visual presentation of the data and information in the figure, making it easier for readers to interpret and understand the content.

**Q: 6) Page 13, Fig. 9. Please explain why the reconstructed products were more blurred than the original product.**

**Response:** Thanks for the comment. After analyzing the data and the reconstruction process, we find several factors that may have contributed to the blurring of the results.

Firstly, the reconstruction algorithm itself might introduce a certain level of smoothing. The reconstruction method involves complex mathematical operations such as interpolation. These operations can average out the details in the original data, resulting in a blurred appearance.

Secondly, the quality of the input data plays a crucial role. The original products generally capture the true characteristics of the target phenomenon with high fidelity. In contrast, the reconstructed product depends on the quality and quantity of the available data for reconstruction. If there are limitations in the data, such as missing values or noisy measurements, the reconstruction algorithm may not be able to fully replicate the details of the original. In our case, although we have taken measures to pre - process and filter out outliers, there may still be some uncertainties and inaccuracies that affect the clarity of the reconstructed product.

In addition to the factors we previously mentioned, there is another significant aspect contributing to the difference in the blurred effect between the reconstructed and original products. We stitch and store the daily raster data for a month as a 3D data (2-D spatial + time), which is subsequently fed into the reconstruction model for learning and training. Monthly averages of VOD are the basis for learning these time-series features, but extreme values tend to be ignored when calculating monthly averages.

This smoothing effect can make the reconstructed products appear more blurred compared to the original product, which retains all the fine - grained details, including those extreme values.

We understand that this is an important consideration, and we are exploring ways to better incorporate extreme value information into our reconstruction process to improve the representativeness of the reconstructed products.

**Q: 7) Page 19, Fig. 16. Discuss the reseason of white pixels over land in VOD_st winter.**

**Response:** Thanks for the comment. Maybe the "reseason" in your question is "reason"? We appreciate your concern and have carefully considered the possible reasons, with a focus on the aspect of original data loss.

Regarding VOD data retrieval, Radio Frequency Interference (RFI) is likely to be a critical factor. In winter, RFI may intensify in certain regions for various reasons. For example, the increased use of electronic heating devices or the operation of communication systems in the same frequency bands as the sensors can render the VOD values unreliable. As a result, these values are removed during data retrieval.

Secondly, the snow and ice cover in winter can distort or attenuate the microwave signals used for VOD measurement. This distortion or attenuation can prevent the sensors from accurately detecting the underlying vegetation, leading to data loss.

Furthermore, low temperatures and other harsh winter weather conditions can impact the calibration of the sensors. Inaccurate calibration can produce unreliable measurement results, which are then discarded, contributing to the loss of data.